# Quantum Coherence from Commensurate Driving with Laser Pulses and Decay

Götz S. Uhrig

Lehrstuhl für Theoretische Physik I, TU Dortmund University
Otto-Hahn Straße 4, D-44221 Dortmund, Germany
goetz.uhrig@tu-dortmund.de

December 4, 2019

## Abstract

Non-equilibrium physics is a particularly fascinating field of current research. Generically, driven systems are gradually heated up so that quantum effects die out. In contrast, we show that a driven central spin model including controlled dissipation in a highly excited state allows us to distill quantum coherent states, indicated by a substantial reduction of entropy; the key resource is the commensurability between the periodicity of the pump pulses and the internal processes. The model is experimentally accessible in purified quantum dots or molecules with unpaired electrons. The potential of preparing and manipulating coherent states by designed driving potentials is pointed out.

# 1 Introduction

Controlling a quantum mechanical system in a coherent way is one of the long-standing goals in physics. Obviously, coherent control is a major ingredient for handling quantum information. In parallel, non-equilibrium physics of quantum systems is continuing to attract significant interest. A key issue in this field is to manipulate systems in time such that their properties can be tuned and changed at will. Ideally, they display properties qualitatively different from what can be observed in equilibrium systems. These current developments illustrate the interest in understanding the dynamics induced by time-dependent Hamiltonians $H(t)$.

The unitary time evolution operator $U(t_2, t_1)$ induced by $H(t)$ is formally given by

$$U(t_2, t_1) = \mathcal{T} \exp\left(-i \int_{t_1}^{t_2} H(t)dt\right) \tag{1}$$

where $\mathcal{T}$ is the time ordering operator. While the explicit calculation of $U(t_2, t_1)$ can be extremely difficult it is obvious that the dynamics induced by a time-dependent Hamiltonian maps quantum states at $t_1$ to quantum states at $t_2$ bijectively and conserves the mutual scalar products. Hence, if initially the system is in a mixed state with high entropy $S > 0$ it stays in a mixed state for ever with exactly the same entropy. No coherence can be generated in this way even for a complete and ideal control of $H(t)$ in time. Hence, one has to consider open systems.

The standard way to generate a single state is to bring the system of interest into thermal contact with a cold system. Generically, this is an extremely slow process. The targeted quantum states have to be ground states of some given system. Alternatively, optical pumping in general and laser cooling in particular [1] are well established techniques to lower the entropy of microscopic systems using resonant pumping and spontaneous decay. Quite recently, engineered dissipation has been recognized as a means to generate targeted entangled quantum states in small [2, 3] and extended systems [4, 5]. Experimentally, entanglement has been shown for two quantum bits [6, 7] and for two trapped mesoscopic cesium clouds [8].

In this article, we show that periodic driving can have a quantum system converge to coherent quantum states if an intermediate, highly excited and decaying state is involved. The key aspect is the commensurability of the periodic pump pulses to the internal process. This distinguishes our proposal from established optical pumping protocols. The completely disordered initial mixture can be made almost coherent. The final mixture only has an entropy $S \approx k_{\mathrm{B}} \ln 2$ corresponding to a mixture of two states. An appealing asset is that once the driving is switched off the Lindbladian decay does not matter anymore and the system is governed by Hamiltonian dynamics only.

The focus of the present work is to exemplarily demonstrate the substantial reduction of entropy in a small spin system subject to periodic laser pulses. The choice of system is motivated by experiments on the electronic spin in quantum dots interacting with nuclear spins [9–16]. The model studied is also applicable to the electronic spin in molecular radicals [17] or to molecular magnets, see Refs. [18–20]. In organic molecules the spin bath is given by the nuclear spins of the hydrogen nuclei in organic ligands.

# 2 Model

The model comprises a central, electronic spin $S = 1/2$ which is coupled to nuclear spins

$$H_{\mathrm{spin}} = H_{\mathrm{CS}} + H_{\mathrm{eZ}} + H_{\mathrm{nZ}} \tag{2}$$

where $H_{\mathrm{eZ}} = hS^x$ is the electronic Zeeman term with $h = g\mu_{\mathrm{B}}B$ ($\hbar$ is set to unity) and the external magnetic field $B$ in $x$-direction; $H_{\mathrm{nZ}} = zh \sum_{i=1}^{N} I_i^x$ is the Zeeman term acting on the nuclear spins taken to be $I = 1/2$. Due to the large nuclear mass, the factor $z$ is of the order of $10^{-3}$, but in principle other $z$-values can be studied as well. In the central spin part $H_{\mathrm{CS}} = \vec{S} \cdot \vec{A}$ the Overhauser field $\vec{A}$ results from the hyperfine interactions $J_i$ between the nuclei and the central spin

$$\vec{A} = \sum_{i=1}^{N} J_i \vec{I}_i. \tag{3}$$

We assume that the couplings $J_i$ are distributed evenly within a certain interval. Besides the spin system there is a single trion state $|\mathrm{T}\rangle$ polarised in $z$-direction at the very high energy $\varepsilon$ ($\approx 1$ eV) so that the total Hamiltonian reads

$$H = H_{\mathrm{spin}} + \varepsilon |\mathrm{T}\rangle\langle\mathrm{T}|. \tag{4}$$

The laser pulse is taken to be very short as in experiment where it is of the order of picoseconds. Hence, we describe it as instantaneous unitary $U_{\mathrm{puls}}$ which takes the $|\uparrow\rangle$ of the central spin to the trion state and vice versa

$$U_{\mathrm{puls}} = c^\dagger + c + |\downarrow\rangle\langle\downarrow|. \tag{5}$$

where $c := |\uparrow\rangle\langle\mathrm{T}|$ and $c^\dagger := |\mathrm{T}\rangle\langle\uparrow|$. Such pulses are applied in long periodic trains lasting seconds and minutes. The repetition time between two consecutive pulses is $T_{\mathrm{rep}}$ of the order of 10 ns.

The decay of the trion is described by the Lindblad equation for the density matrix $\rho$

$$\partial_t \rho(t) = -i[H, \rho] - \gamma(c^\dagger c \rho + \rho c^\dagger c - 2c\rho c^\dagger) \tag{6}$$

where the prefactor $\gamma > 0$ of the dissipator term [21] defines the decay rate. The corresponding process with $c$ and $c^\dagger$ swapped needs not be included because its decay rate is smaller by $\exp(-\beta\varepsilon)$, i.e., it vanishes for all physical purposes.

## 3 Mathematical Properties of Time Evolution

The key observation is that the dynamics from just before the $n$th pulse at $t = nT_{\mathrm{rep}}-$ to just before the $n + 1$st pulse at $t = (n + 1)T_{\mathrm{rep}}-$ is a *linear* mapping $M : \rho(nT_{\mathrm{rep}}-) \to \rho((n+1)T_{\mathrm{rep}}-)$ which does not depend on $n$. Since it is acting on operators one may call it a superoperator. Its matrix form is derived explicitly in Appendix A. If no dissipation took place ($\gamma = 0$) the mapping $M$ would be unitary. But in presence of the dissipative trion decay it is a general matrix with the following properties:

1. The matrix $M$ has an eigenvalue 1 which may be degenerate. If the dynamics of the system takes place in $n$ separate subspaces without transitions between them the degeneracy is at least $n$.

2. All eigenoperators to eigenvalues different from 1 are traceless.

3. At least one eigenoperator to eigenvalue 1 has a finite trace.

4. The absolute values of all eigenvalues of $M$ are not larger than 1.

5. If there is a non-real eigenvalue $\lambda$ with eigenoperator $C$, the complex conjugate $\lambda^*$ is also an eigenvalue with eigenoperator $C^\dagger$.

6. The eigenoperators to eigenvalues 1 can be scaled to be hermitian.

While the above properties can be shown rigorously, see Appendix B, for any Lindblad evolution, the following ones are observed numerically in the analysis of the particular model (6) under study here:

(a) The matrix $M$ is diagonalizable; it does not require a Jordan normal form.

(b) For pairwise different couplings $i \neq j \Rightarrow J_i \neq J_j$ the eigenvalue 1 is non-degenerate.

(c) The eigenoperators to eigenvalue 1 can be scaled to be hermitian and non-negative. In the generic, non-degenerate case we denote the properly scaled eigenoperator $V_0$ with $\mathrm{Tr}(V_0) = 1$.

(d) No eigenvalue $\neq 1$, but with absolute value 1, occurs, i.e., all eigenvalues different from 1 are smaller than 1 in absolute value.

(e) Complex eigenvalues and complex eigenoperators do occur.

The above properties allow us to understand what happens in experiment upon application of long trains of pulses corresponding to $10^{10}$ and more applications of $M$. Then it is safe to conclude that all contributions from eigenoperators to eigenvalues smaller than 1 have died out completely. Only the (generically) single eigenoperator $V_0$ to eigenvalue 1 is left such that

$$\lim_{n \to \infty} \rho(nT_{\mathrm{rep}}-) = V_0. \tag{7}$$

The quasi-stationary state after long trains of pulses is given by $V_0$ [1]. This observation simplifies the calculation of the long-time limit greatly compared to previous quantum mechanical studies [12,13,16,22]. One has to compute the eigenoperator of $M$ to the eigenvalue 1. Below this is performed by diagonalization of $M$ which is a reliable approach, but restricted to small systems $N \lesssim 6$. We stress that no complete diagonalization is required to know $V_0$ because only the eigenoperator to the eigenvalue 1 is needed. Hence we are optimistic that further computational improvements are possible. If, however, the speed of convergence is of interest more information on the spectrum and the eigenoperators of $M$ is needed, see also Sect. 5.

## 4 Results on Entropy

It is known that in pulsed quantum dots nuclear frequency focusing occurs (NFF) [9,10,23] which can be explained by a significant change in the distribution of the Overhauser field [11–13, 15, 16, 22] which is Gaussian initially. This distribution develops a comb structure with equidistant spikes. The difference $\Delta A_x$ between consecutive spikes is such that it corresponds to a full additional revolution of the central spin $T_{\mathrm{rep}}\Delta A_x = 2\pi$. A comb-like probability distribution is more structured and contains more information than the initial featureless Gaussian. For instance, the entropy reduction of the Overhauser field distributions computed in Ref. [16], Fig. 12, relative to the initial Gaussians is $\Delta S = -0.202k_{\mathrm{B}}$ at $B = 0.93\mathrm{T}$ and $\Delta S = -0.018k_{\mathrm{B}}$ at $B = 3.71\mathrm{T}$. Hence, NFF decreases the entropy, but only slightly for large spin baths. This observation inspires us to ask to which extent continued pulsing can reduce entropy and which characteristics the final state has.

---

[1]We use the term 'quasi-stationary' state because it is stationary only if we detect it stroboscopically at the time instants $t = nT_{\mathrm{rep}}-$.

Inspired by the laser experiments on quantum dots [9, 10, 23] we choose an (arbitrary) energy unit $J_Q$ and thus $\hbar/J_Q$ as time unit which can be assumed to be of the order of 1ns. The repetition time $T_{rep}$ is set to $4\pi\hbar/J_Q$ which is on the one hand close to the experimental values where $T_{rep} = 13.2$ns and on the other hand makes it easy to recognize resonances, see below. The trion decay rate is set to $2\gamma = 2.5J_Q$ to reflect a trion life time of $\approx 0.4$ps. The bath size is restricted to $N \in \{1, 2, \ldots, 6\}$, but still allows us to draw fundamental conclusions and to describe electronic spins coupled to hydrogen nuclear spins in small molecules [17–20]. The individual couplings $J_i$ are chosen to be distributed according to

$$J_i = J_{max}(\sqrt{5} - 2)\left(\sqrt{5} + 2(i-1)/(N-1)\right), \tag{8}$$

which is a uniform distribution between $J_{min}$ and $J_{max}$ with $\sqrt{5}$ inserted to avoid accidental commensurabilities. Results for a frequently used exponential parameterization [24]

$$J_i = J_{max}\exp(-\alpha(i-1)/(N-1)) \tag{9}$$

with $\alpha \in \{0.5, 1\}$ and for a Gaussian parametrization motivated by the electronic wave function in quantum dots [25]

$$J_i = J_{max}\exp(-\alpha[(i-1)/(N-1)]^2). \tag{10}$$

are given in the next section and in Appendix D.

Figure 1 displays a generic dependence on the external magnetic field $h = g\mu_B B_x$ of the entropy of the limiting density matrix $V_0$ obtained after infinite number of pulses. Two nested resonances of the Larmor precessions are discernible: the central electronic spin resonates for $hT_{rep} = 2\pi n$ ($n \in \mathbb{Z}$) while the nuclear bath spin resonates for $zhT_{rep} = 2\pi n'$. These conditions, however, apply only without pulsing or interactions. The driven system displays important shifts. The nuclear resonance appears to be shifted by $z\Delta h \approx \pm J_{max}/2$, see right panel of Fig. 1(a). The explanation is that the dynamics of the central spin $S = 1/2$ creates an additional magnetic field acting on each nuclear spin of the order of $J_i/2$ which is estimated by $J_{max}/2$. Further support of this explanation is given in Appendix C.

The electronic resonance is shifted by

$$\Delta h = \pm A_{max} \tag{11}$$

where $A_{max}$ is the maximum Overhauser field given by $A_{max} := (1/2)\sum_{i=1}^N J_i$ for maximally polarized bath spins. This is shown in the right panel of Fig. 1(b). The commensurability of these resonances is crucial for the advocated mechanism.

For the minimum entropy, it is essential not to look at the bare resonances, but to take the two above mentioned shifts into account. We posed the question to which extent the initial entropy of complete disorder $S_{init} = k_B(N+1)\ln 2$ (in the figures and henceforth $k_B$ is set to unity) can be reduced by commensurate periodic pumping. The results in Fig. 1 clearly show that remarkably low values of entropy can be reached. The residual value of $S \approx 0.5k_B$ in the minima of the right panel of Fig. 1(b) corresponds to a contribution of less than two states ($S = \ln 2k_B \approx 0.7k_B$) while initially 16 states were mixed for $N = 3$ so that the initial entropy is $S_{init} = 4\ln 2k_B \approx 2.77k_B$. This represents a remarkable distillation of coherence.

Hence, we focus on the minima and in particular on the left minimum. We address the question whether the distillation of coherence still works for larger systems. Unfortunately, the numerical analysis cannot be extended easily due to the dramatically increasing dimension $D = 2^{2(N+1)}$ because we are dealing with the Hilbert space of density matrices

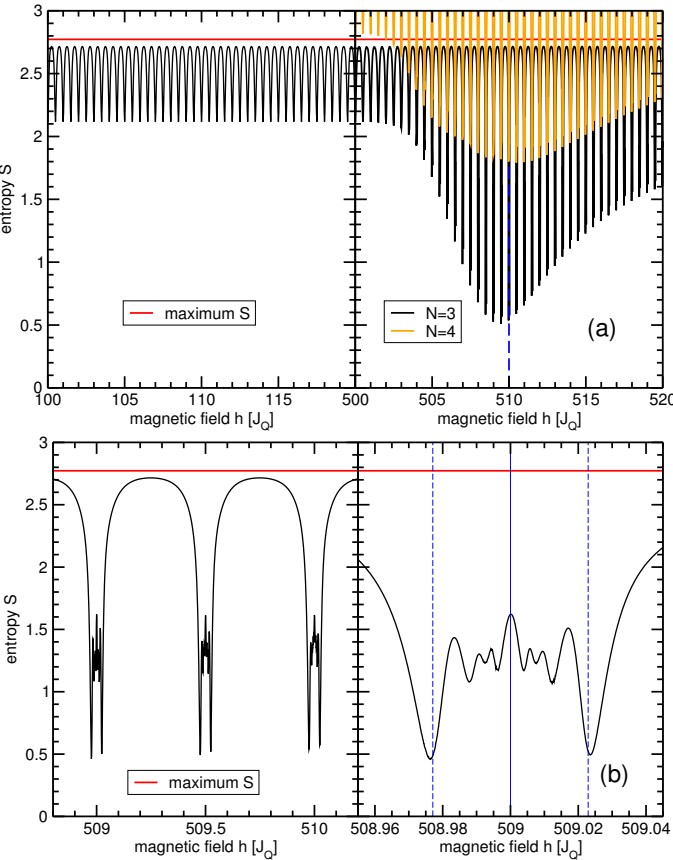

Figure 1: (a) Residual entropy of the limiting density matrix $V_0$ obtained after infinite number of pulses vs. the applied magnetic field for $J_{\max} = 0.02 J_Q$ and $z = 1/1000$; 1 Tesla corresponds roughly to $50 J_Q$. Resonances of the electronic spin occur every $\Delta h = 0.5 J_{\max}$; resonances of the nuclear spins occur every $\Delta h = 500 J_{\max}$. The blue dashed line depicts an offset of $\Delta h = \pm J_{\max}/(2z)$ from the nuclear resonance. (b) Zooms into intervals of the magnetic field where the lowest entropies are reached. The blue dashed lines depict an offset of $\Delta h = \pm A_{\max}$ from the electronic resonance.

of the spin bath and the central spin. Yet a trend can be deduced from results up to $N = 6$ displayed in Fig. 2(a). The entropy reduction per $N + 1$ spins is $-0.58 k_B$ for $N = 3$, $-0.57 k_B$ for $N = 4$, $-0.55 k_B$ for $N = 5$, and $-0.52 k_B$ for $N = 6$. The reduction is substantial, but slowly decreases with system size. Presently, we cannot know the behavior for $N \to \infty$. The finite value $\approx -0.2 k_B$ found in the semiclassical simulation [15, 16] indicates that the effect persists for large baths. In Appendix D, results for the couplings defined in (9) or in (10) are given which corroborate our finding. The couplings may be rather close to each other, but not equal. It appears favorable that the spread of couplings is not too large.

Which state is reached in the minimum of the residual entropy? The decisive clue is provided by the lower panel Fig. 2(b) displaying the polarization of the spin bath. It is normalized such that its saturation value is unity. Clearly, the minimum of the residual entropy coincides with the maximum of the polarization. The latter is close to its saturation value though not quite with a minute decrease for increasing $N$. This tells us that the limiting density matrix $V_0$ essentially corresponds to the polarized spin bath. The central electronic spin is also almost perfectly polarized (not shown), but in $z$-direction. These observations clarify the state which can be retrieved by long trains of pulses.

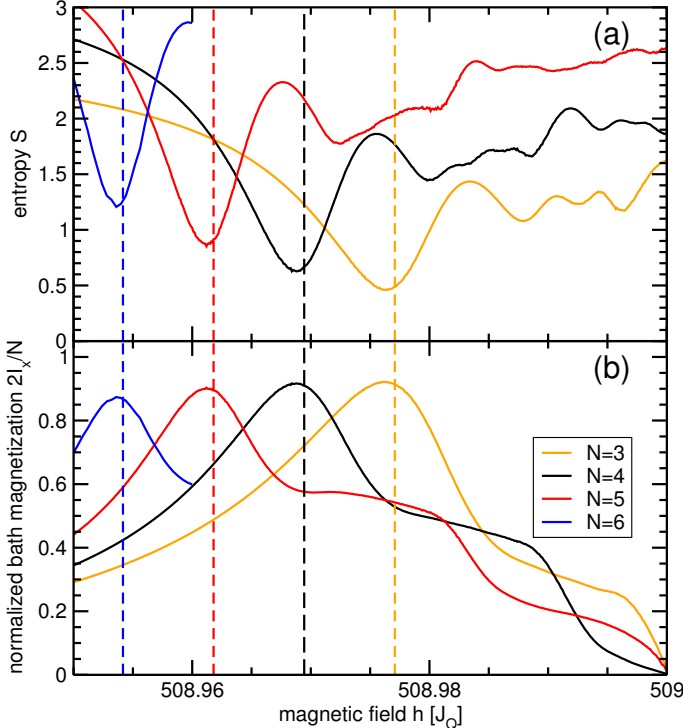

Figure 2: (a) Residual entropy of the limiting density matrix $V_0$ for various bath sizes; other parameters as in Fig. 1. The dashed lines indicate the shifts of the electronic resonance by $-A_{\max}$. (b) Corresponding normalized polarization of the spin bath in the external field direction, i.e. the $x$-direction.

Additionally, Fig. 2(b) explains the shift of the electronic resonance. The polarized spin bath renormalizes the external magnetic field by (almost) $\pm A_{\max}$. To the left of the resonance, it enhances the external field $(+A_{\max})$ while the external field is effectively reduced $(-A_{\max})$ to the right of the resonance. Note that an analogous direct explanation for the shift of the nuclear resonance in the right panel of Fig. 1 is not valid. The computed polarization of the central spin points in $z$-direction and thus does not shift the external field.

## 5 Results on Convergence

In order to assess the speed of convergence of the initially disordered density matrix $\rho_0 = \mathbb{1}/Z$ to the limiting density matrix $V_0$ we proceed as follows. Let us assume that the matrices $v_i$ are the eigen matrices of $M$ and that they are normalized $||v_i||^2 := \mathrm{Tr}(v_i^\dagger v_i) = 1$. Since the mapping $M$ is not unitary, orthogonality of the eigenmatrices cannot be assumed. Note that the standard normalization generically implies that there is some factor between $V_0$ with $\mathrm{Tr}(V_0) = 1$ and $v_0$. The initial density matrix $\rho_0$ can be expanded in the $\{v_i\}$

$$\rho_0 = \sum_{j=0}^{D-1} \alpha_j v_j. \tag{12}$$

After $n$ pulses, the density matrix $\rho_n$ is given by

$$\rho_n = \sum_{j=0}^{D-1} \alpha_j \lambda_j^n v_j \tag{13}$$

where $\lambda_j$ are the corresponding eigenvalues of $M$ and $\lambda_0 = 1$ by construction. We aim at $\rho_0$ being close to $V_0$ within $p_{\text{thresh}}$, i.e.,

$$||\rho_n - V_0|| \leq p_{\text{thresh}}||V_0|| \tag{14}$$

should hold for an appropriate $n$. A generic value of the threshold $p_{\text{thresh}}$ is 1%. To this end, the minimum $n$ which fulfills (14) has to be estimated.

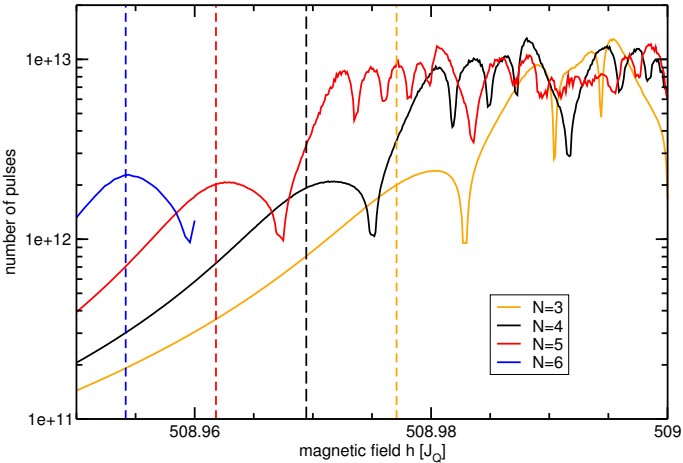

Figure 3: Number of pulses for a convergence within 1% ($p_{\text{thresh}} = 0.0$) are plotted for various bath sizes; couplings given by (8), other parameters as in Fig. 1. The corresponding residual entropies and magnetizations are depicted in Fig. 2. The vertical dashed lines indicate the estimates (11) for the entropy minima as before.

Such an estimate can be obtained by determining

$$n_j := 1 + \text{trunc} \left[ \frac{\ln(|p_{\text{thresh}} \alpha_0 / \alpha_j|)}{\ln(|\lambda_j|)} \right] \tag{15}$$

for $j \in \{1, 2, 3, \ldots, D-1\}$. The estimate of the required number of pulses is the maximum of these number, i.e.,

$$n_{\text{puls}} := \max_{1 \leq j < D} n_j. \tag{16}$$

We checked exemplarily that the number determined in this way implies that the convergence condition (14) is fulfilled. This is not mathematically rigorous because it could be that there are very many slowly decreasing contributions which add up to a significant deviation from $V_0$. But generically, this is not the case.

In Fig. 3 the results are shown for various bath sizes and the parameters for which the data of the previous figures was computed. Since the entropy minima are located at the positions of the vertical dashed lines to good accuracy one can read off the required number of pulses at the intersections of the solid and the dashed lines. Clearly, about $2 \cdot 10^{12}$ pulses are necessary to approach the limiting, relatively pure density matrices $V_0$. Interestingly, the number of required pulses does not depend much on the bath size, at least for the accessible bath sizes. This is a positive message in view of the scaling towards larger baths in experimental setups.

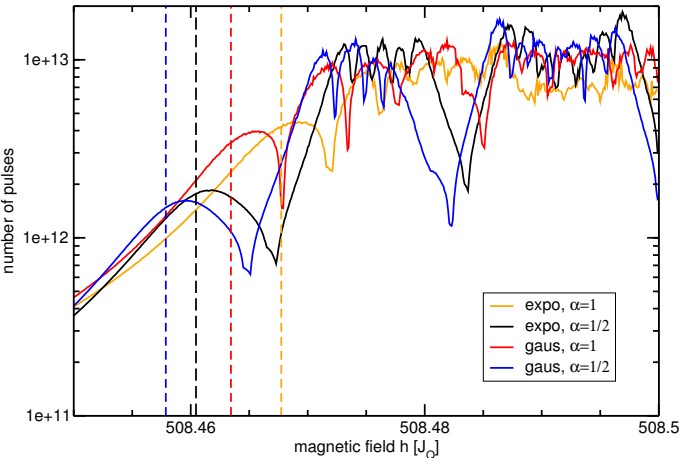

Figure 4: Number of pulses for a convergence within 1% ($p_{\text{thresh}} = 0.01$) for $N = 5$, $J_{\text{max}} = 0.02 J_{\text{Q}}$, and $z = 10^{-3}$ for the exponential parametrization in (9) (legend "expo") and the Gaussian parametrization in (10) (legend "gaus"). The corresponding residual entropies and magnetizations are depicted in Figs. 8 and 9, respectively. The vertical dashed lines indicate the estimates for the entropy minima which are shifted from the resonances without interactions according to (11).

Figure 4 depicts the required minimum number of pulses for the two alternative parametrizations of the couplings (9) and (10). Again, the range is about $3 \cdot 10^{12}$. Still, there are relevant differences. The value $n_{\text{puls}}$ is higher for $\alpha = 1$ ($\approx 4 \cdot 10^{12}$) than for $\alpha = 1/2$ ($\lesssim 2 \cdot 10^{12}$). This indicates that the mechanism of distilling quantum states by commensurability with periodic external pulses works best if the couplings are similar, i.e., if their spread is small. The same qualitative result was obtained for the residual entropy, see Appendix D. Note that this argument also explains why the Gaussian parametrized couplings (10) require slightly less pulses than the exponential parametrized couplings (9). One could have thought that the cumulated couplings $J_i \approx J_{\text{max}}$ in the Gaussian case require longer pulsing in order to achieve a given degree of distillation because mathematically equal couplings $J_i = J_{i'}$ imply degeneracies preventing distillation, see the mathematical properties discussed in Sect. 3. But this is obviously not the case.

The total numbers of pulses is rather high. As many as $2 \cdot 10^{12}$ pulses for a repetition time $T_{\text{rep}} \approx 10$ns imply about six hours of pulsing. This can be achieved in the lab, but the risk that so far neglected decoherence mechanisms spoil the process is real. If, however, the pulses can be applied more frequently, for instance with $T_{\text{rep}} = 1$ns, the required duration shrinks to about 30 minutes. The question arises why so many pulses are required. While a comprehensive study of this aspect is beyond the scope of the present article, first clue can be given.

It suggests itself that the slow dynamics in the bath is responsible for the large number of pulses required for convergence. This idea is corroborated by the results displayed in Fig. 5 where a larger maximum coupling and, importantly, a larger $z$ factor is assumed. Recall that the $z$-factor is the ratio of the Larmor frequency of the bath spins to the Larmor frequency of the central spin. If it is increased, here by a factor of 100, the bath spins precess much quicker. Indeed, the range of the required number of pulses is much lower with $2 \cdot 10^7$ which is five orders of magnitude less than for the previous parameters. The former six hours then become fractions of seconds. Of course, the conventional $g$-factors of nuclear and electronic spins do not allow for $z = 0.1$. But the central spin model as such, built by a central spin and a bath of spins supplemented by a damped excitation

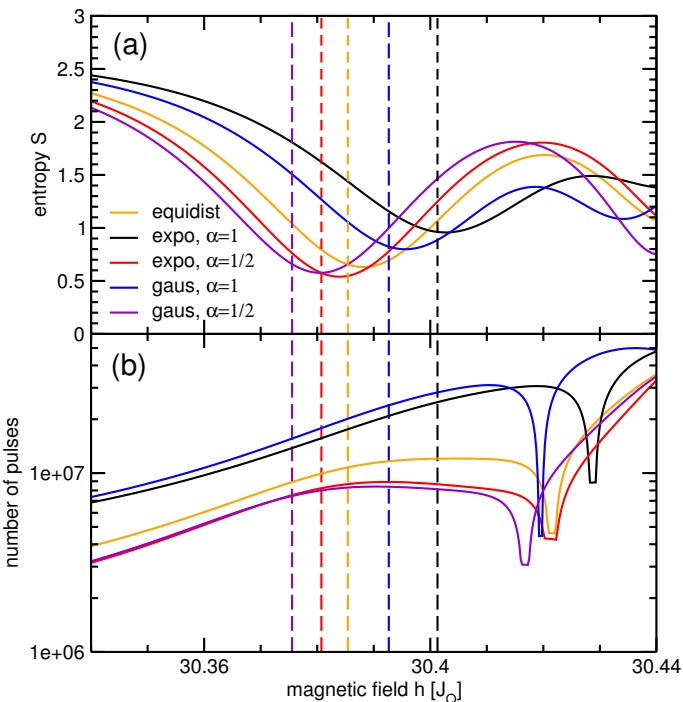

Figure 5: Residual entropies (panel a) and number of pulses (panel b) for a convergence within 1% ($p_{\text{thresh}} = 0.0$) for $N = 3$, $J_{\max} = 0.1J_{\text{Q}}$, and $z = 0.1$ for the equidistant parametrization in (8) (legend "equidist"), the exponential parametrization in (9) (legend "expo") and the Gaussian parametrization in (10) (legend "gaus"). The vertical dashed lines indicate the estimates for the entropy minima which are shifted from the resonances without interactions according to (11).

can also be realized in a different physical system.

## 6    Conclusion

Previous work has established dynamic nuclear polarization (DNP), for a review see Ref. [26]. But it must be stressed that the mechanism of this conventional DNP is fundamentally different from the one described here. Conventionally, the polarization of an electron is transferred to the nuclear spins, i.e., the polarization of the electrons induces polarization of the nuclei in the *same* direction. In contrast, in the setup studied here, the electron is polarized in $z$-direction while the nuclear spins are eventually polarized perpendicularly in $x$-direction. Hence, the mechanism is fundamentally different: it is NFF stemming essentially from commensurability. This is also the distinguishing feature compared to standard optical pumping. States in the initial mixture which do not allow for a time evolution commensurate with the repetition time $T_{\text{rep}}$ of the pulses are gradually suppressed. Eventually, only the particular state which allows for a dynamics commensurate with $T_{\text{rep}}$ persists. This mechanism can be used also for completely different physical systems, e.g., in ensembles of oscillators. The studied case of coupled spins extends the experimental and theoretical observations of NFF for large spin baths [9–16] where many values of the polarization of the Overhauser field can lead to commensurate dynamics. Hence, only a partial reduction of entropy occurred.

   The above established DNP by NFF comprises the potential for a novel experimental

technique for state preparation: laser pulses instead of microwave pulses as in standard NMR can be employed to prepare coherent states which can be used for further processing, either to perform certain quantum protocols or for analysis of the systems under study. The combination of optical and radio frequency pulsing appears promising because it enlarges the possibilities of experimental manipulations. Another interesting perspective is to employ the concept of state distillation by commensurability to physical systems other than localized spins, for instance to spin waves in quantum magnets. A first experimental observations of commensurability effects for spin waves in ferromagnets are already carried out [27].

In summary, we showed that dissipative dynamics of a highly excited state is sufficient to modify the dynamics of energetically low-lying spin degrees of freedom away from unitarity. The resulting dynamic map acts like a contraction converging towards a single density matrix upon iterated application. The crucial additional ingredient is *commensurability* which enables a substantial entropy reduction, almost to a single pure state. This has been explicitly shown for an exemplary small central spin model including electronic and nuclear Zeeman effect. This model served as proof-of-principle model to establish the mechanism of distillation by commensurability.

Such a model describes the electronic spin in quantum dots with diluted nuclear spin bath or the spin of unpaired electrons in molecules, hyperfine coupled to nuclear hydrogen spins. We stress that the mechanism of commensurability can also be put to use in other systems with periodic internal processes. The fascinating potential to create and to manipulate coherent quantum states by such approaches deserves further investigation.

## Acknowledgements

The author thanks A. Greilich, J. Schnack, and O. P. Sushkov for useful discussions and the School of Physics of the University of New South Wales for its hospitality.

**Funding information** This work was supported by the Deutsche Forschungsgemeinschaft (DFG) and the Russian Foundation of Basic Research in TRR 160, by the DFG in project no. UH 90-13/1, and by the Heinrich-Hertz Foundation of Northrhine-Westfalia.

## A  Derivation of the Linear Mapping

The goal is to solve the time evolution of $\rho(t)$ from just before a pulse until just before the next pulse. Since the pulse leads to a unitary time evolution which is linear

$$\rho(nT_{\mathrm{rep}}-) \to \rho(nT_{\mathrm{rep}}+) = U_{\mathrm{puls}}\rho(nT_{\mathrm{rep}}-)U_{\mathrm{puls}}^{\dagger} \tag{17}$$

with $U_{\mathrm{puls}}$ from (5) and the subsequent Lindblad dynamics defined by the linear differential equation (6) is linear as well the total propagation in time is given by a linear mapping $M : \rho(nT_{\mathrm{rep}}-) \to \rho((n+1)T_{\mathrm{rep}}-)$. This mapping is derived here by an extension of the approach used in Ref. [16].

The total density matrix acts on the Hilbert space given by the direct product of the Hilbert space of the central spin comprising three states $(|\uparrow\rangle, |\downarrow\rangle, |\mathrm{T}\rangle)$ and the Hilbert space of the spin bath. We focus on $\rho_{\mathrm{TT}} := \langle \mathrm{T}|\rho|\mathrm{T}\rangle$ which is a $2^N \times 2^N$ dimensional density matrix for the spin bath alone because the central degree of freedom is traced out. By $\rho_{\mathrm{S}}$ we denote the $d \times d$ dimensional density matrix of the spin bath and the central spin, i.e., $d = 2^{N+1}$ since no trion is present: $\rho_{\mathrm{S}}|\mathrm{T}\rangle = 0$. The number of entries in the

density matrix is $D = d^2$, i.e., the mapping we are looking for can be represented by a $D \times D$ matrix.

The time interval $T_{\mathrm{rep}}$ between two consecutive pulses is sufficiently long so that all excited trions have decayed before the next pulse arrives. In numbers, this means $2\gamma T_{\mathrm{rep}} \gg 1$ and implies that $\rho(nT_{\mathrm{rep}}-) = \rho_{\mathrm{S}}(nT_{\mathrm{rep}}-)$ and hence inserting the unitary of the pulse (5) yields

$$\rho(nT_{\mathrm{rep}}+) = U_{\mathrm{puls}}\rho_{\mathrm{S}}(nT_{\mathrm{rep}}-)U_{\mathrm{puls}}^\dagger \tag{18a}$$

$$\rho_{\mathrm{TT}}(nT_{\mathrm{rep}}+) = \langle\uparrow|\rho_{\mathrm{S}}(nT_{\mathrm{rep}}-)|\uparrow\rangle \tag{18b}$$

$$\rho_{\mathrm{S}}(nT_{\mathrm{rep}}+) = |\downarrow\rangle\langle\downarrow|\rho_{\mathrm{S}}(nT_{\mathrm{rep}}-)|\downarrow\rangle\langle\downarrow| = S^-S^+\rho_{\mathrm{S}}(nT_{\mathrm{rep}}-)S^-S^+ \tag{18c}$$

where we used the standard ladder operators $S^\pm$ of the central spin to express the projection $|\downarrow\rangle\langle\downarrow|$. The equations (18) set the initial values for the subsequent Lindbladian dynamics which we derive next. For completeness, we point out that there are also non-diagonal contributions of the type $\langle\mathrm{T}|\rho|\uparrow\rangle$, but they do not matter for $M$.

Inserting $\rho_{\mathrm{TT}}$ into the Lindblad equation (6) yields

$$\partial_t\rho_{\mathrm{TT}}(t) = -i[H_{\mathrm{nZ}}, \rho_{\mathrm{TT}}(t)] - 2\gamma\rho_{\mathrm{TT}}(t). \tag{19}$$

No other parts contribute. The solution of (19) reads

$$\rho_{\mathrm{TT}}(t) = e^{-2\gamma t}e^{-iH_{\mathrm{nZ}}t}\rho_{\mathrm{TT}}(0+)e^{iH_{\mathrm{nZ}}t}. \tag{20}$$

By the argument $0+$ we denote that the initial density matrix for the Lindbladian dynamics is the one just after the pulse.

For $\rho_{\mathrm{S}}$, the Lindblad equation (6) implies

$$\partial_t\rho_{\mathrm{S}}(t) = -i[H_{\mathrm{spin}}, \rho_{\mathrm{S}}(t)] + 2\gamma|\uparrow\rangle\rho_{\mathrm{TT}}(t)\langle\uparrow|. \tag{21}$$

Since we know the last term already from its solution in (20) we can treat it as given inhomogeneity in the otherwise homogeneous differential equation. With the definition $U_{\mathrm{S}}(t) := \exp(-iH_{\mathrm{spin}}t)$ we can write

$$\partial_t\left(U_{\mathrm{S}}^\dagger(t)\rho_{\mathrm{S}}(t)U_{\mathrm{S}}(t)\right) = 2\gamma U_{\mathrm{S}}^\dagger(t)|\uparrow\rangle\rho_{\mathrm{TT}}(t)\langle\uparrow|U_{\mathrm{S}}(t). \tag{22}$$

Integration leads to the explicit solution

$$\rho_{\mathrm{S}}(t) = U_{\mathrm{S}}(t)\rho_{\mathrm{S}}(0+)U_{\mathrm{S}}^\dagger(t) + 2\gamma\int_0^t U_{\mathrm{S}}^\dagger(t-t')|\uparrow\rangle\rho_{\mathrm{TT}}(t')\langle\uparrow|U_{\mathrm{S}}(t-t')dt'. \tag{23}$$

If we insert (20) into the above equation we encounter the expression

$$|\uparrow\rangle\exp(-iH_{\mathrm{nZ}}t) = \exp(-iH_{\mathrm{nZ}}t)|\uparrow\rangle = \exp(-izhI_{\mathrm{tot}}^x t)\exp(izhS^x t)|\uparrow\rangle. \tag{24}$$

where $I_{\mathrm{tot}}^x := S^x + \sum_{i=1}^N I_i^x$ is the total momentum in $x$-direction. It is a conserved quantity commuting with $H_{\mathrm{spin}}$ so that a joint eigenbasis with eigenvalues $m_\alpha$ and $E_\alpha$ exists. We determine such a basis $\{|\alpha\rangle\}$ by diagonalization in the $d$-dimensional Hilbert space ($d = 2^{N+1}$) of central spin and spin bath and convert (23) in terms of the matrix elements of the involved operators. For brevity, we write $\rho_{\alpha\beta}$ for the matrix elements of $\rho_{\mathrm{S}}$.

$$\rho_{\alpha\beta}(t) = e^{-i(E_\alpha - E_\beta)t}\Big\{\rho_{\alpha\beta}(0+)$$

$$+ 2\gamma\int_0^t e^{i(E_\alpha - E_\beta - zh(m_\alpha - m_\beta))t'}\langle\alpha|e^{izhS^x t'}|\uparrow\rangle\rho_{\mathrm{TT}}(0+)\langle\uparrow|e^{izhS^x t'}|\beta\rangle dt'\Big\}. \tag{25}$$

Elementary quantum mechanics tells us that

$$e^{izhS^x t'} | \uparrow \rangle = \frac{1}{2} e^{ia} (| \uparrow \rangle + | \downarrow \rangle) + \frac{1}{2} e^{-ia} (| \uparrow \rangle - | \downarrow \rangle) \tag{26}$$

with $a := zht'/2$ which we need for the last row of equation (25). Replacing $\rho_{\mathrm{TT}}(0+)$ by $\langle \uparrow | \rho_{\mathrm{S}}(nT_{\mathrm{rep}}-) | \uparrow \rangle$ according to (18b) and inserting (26) we obtain

$$\langle \alpha | e^{izhS^x t'} | \uparrow \rangle \rho_{\mathrm{TT}}(0+) \langle \uparrow | e^{izhS^x t'} | \beta \rangle = \langle \alpha | e^{izhS^x t'} | \uparrow \rangle \langle \uparrow | \rho_{\mathrm{S}}(0-) | \uparrow \rangle \langle \uparrow | e^{izhS^x t'} | \beta \rangle \tag{27a}$$

$$= \frac{1}{2} \left( R^{(0)} + e^{izht'} R^{(1)} + e^{-izht'} R^{(-1)} \right)_{\alpha\beta} \tag{27b}$$

with the three $d \times d$ matrices

$$R^{(0)} := S^+ S^- \rho_{\mathrm{S}}(0-) S^+ S^- + S^- \rho_{\mathrm{S}}(0-) S^+ \tag{28a}$$

$$R^{(1)} := \frac{1}{2} (S^+ + \mathbb{1}_d) S^- \rho_{\mathrm{S}}(0-) S^+ (S^- - \mathbb{1}_d) \tag{28b}$$

$$R^{(-1)} := \frac{1}{2} (S^+ - \mathbb{1}_d) S^- \rho_{\mathrm{S}}(0-) S^+ (S^- + \mathbb{1}_d). \tag{28c}$$

In this derivation, we expressed ket-bra combinations by the spin ladder operators according to

$$| \uparrow \rangle \langle \uparrow | = S^+ S^- \qquad | \uparrow \rangle \langle \downarrow | = S^+ \qquad | \downarrow \rangle \langle \uparrow | = S^-. \tag{29}$$

The final step consists in inserting (27b) into (25) and integrating the exponential time dependence straightforwardly from 0 to $T_{\mathrm{rep}}$. Since we assume that $2\gamma T_{\mathrm{rep}} \gg 1$ so that no trions are present once the next pulse arrives the upper integration limit $T_{\mathrm{rep}}$ can safely and consistently be replaced by $\infty$. This makes the expressions

$$G_{\alpha\beta}(\tau) := \frac{\gamma}{2\gamma - i[E_\alpha - E_\beta + zh(m_\beta - m_\alpha + \tau)]} \tag{30}$$

appear where $\tau \in \{-1, 0, 1\}$. Finally, we use (18c) and summarize

$$\rho_{\alpha\beta}(t) = e^{-i(E_\alpha - E_\beta)t} \left\{ (S^- S^+ \rho_{\mathrm{S}}(0-) S^- S^+)_{\alpha\beta} + \sum_{\tau=-1}^{1} G_{\alpha\beta}(\tau) R_{\alpha\beta}^{(\tau)} \right\}. \tag{31}$$

This provides the complete solution for the dynamics of $d \times d$ matrix $\rho_{\mathrm{S}}$ from just before a pulse ($t = 0-$) till just before the next pulse for which we set $t = T_{\mathrm{rep}}$ in (31).

In order to set up the linear mapping $M$ as $D \times D$ dimensional matrix with $D = d^2$ we denote the matrix elements $M_{\mu'\mu}$ where $\mu$ is a combined index for the index pair $\alpha\beta$ and $\mu'$ for $\alpha'\beta'$ with $\alpha, \beta, \alpha', \beta' \in \{1, 2 \ldots, d\}$. For brevity, we introduce

$$P_{\alpha\beta} := [(S^+ + \mathbb{1}_d) S^-]_{\alpha\beta} \qquad Q_{\alpha\beta} := [(S^+ - \mathbb{1}_d) S^-]_{\alpha\beta}. \tag{32}$$

Then, (31) implies

$$M_{\mu'\mu} = \frac{1}{2} e^{-i(E_{\alpha'} - E_{\beta'})T_{\mathrm{rep}}} \left\{ 2(S^- S^+)_{\alpha'\alpha} (S^- S^+)_{\beta\beta'} \right.$$

$$+ 2 G_{\alpha'\beta'}(0) \left[ (S^+ S^-)_{\alpha'\alpha} (S^+ S^-)_{\beta\beta'} + S_{\alpha'\alpha}^- S_{\beta\beta'}^+ \right]$$

$$\left. + \left[ G_{\alpha'\beta'}(1) P_{\alpha'\alpha} Q_{\beta'\beta}^* + G_{\alpha'\beta'}(-1) Q_{\alpha'\alpha} P_{\beta'\beta}^* \right] \right\}. \tag{33}$$

This concludes the explicit derivation of the matrix elements of $M$. Note that they are relatively simple in the sense that no sums over matrix indices are required on the right hand side of (33). This relative simplicity is achieved because we chose to work in the eigenbasis of $H_{\mathrm{spin}}$. Other choices of basis are possible, but render the explicit respresentation significantly more complicated.

# B   Properties of the Time Evolution

**Preliminaries**   Here we state several mathematical properties of the mapping $M$ which hold for any Lindblad dynamics over a given time interval which can be iterated arbitrarily many times. We assume that the underlying Hilbert space is $d$ dimensional so that $M$ acts on the $D = d^2$ dimensional Hilbert space of $d \times d$ matrices, i.e., $M$ can be seen as $D \times D$ matrix. We denote the standard scalar product in the space of operators by

$$(A|B) := \mathrm{Tr}(A^\dagger B) \tag{34}$$

where the trace refers to the $d \times d$ matrices $A$ and $B$.

Since no state of the physical system vanishes in its temporal evolution $M$ conserves the trace of any density matrix

$$\mathrm{Tr}(M\rho) = \mathrm{Tr}(\rho). \tag{35}$$

This implies that $M$ conserves the trace of *any* operator $C$. This can be seen by writing $C = (C + C^\dagger)/2 + (C - C^\dagger)/2 = R + iG$ where $R$ and $G$ are hermitian operators. They can be diagonalized and split into their positive and their negative part $R = p_1 - p_2$ and $G = p_3 - p_4$. Hence, each $p_i$ is a density matrix up to some real, positive scaling and we have

$$C = p_1 - p_2 + i(p_3 - p_4). \tag{36}$$

Then we conclude

$$\mathrm{Tr}(MC) = \mathrm{Tr}(Mp_1) - \mathrm{Tr}(Mp_2) + i(\mathrm{Tr}(Mp_3) - \mathrm{Tr}(Mp_4)) \tag{37a}$$

$$= \mathrm{Tr}(p_1) - \mathrm{Tr}(p_2) + i(\mathrm{Tr}(p_3) - \mathrm{Tr}(p_4)) \ = \ \mathrm{Tr}(C). \tag{37b}$$

**Property 1.**   The conservation of the trace for any $C$ implies

$$\mathrm{Tr}(C) = (\mathbb{1}_d|C) = (\mathbb{1}_d|MC) = (M^\dagger \mathbb{1}_d|C) \tag{38}$$

where $\mathbb{1}_d$ is the $d \times d$-dimensional identity matrix and $M^\dagger$ is the $D \times D$ hermitian conjugate of $M$. From (38) we conclude

$$M^\dagger \mathbb{1}_d = \mathbb{1}_d \tag{39}$$

which means that $\mathbb{1}_d$ is an eigenoperator of $M^\dagger$ with eigenvalue 1. Since the characteristic polynomial of $M$ is the same as the one of $M^\dagger$ up to complex conjugation we immediately see that 1 is also an eigenvalue of $M$. If the dynamics of the system takes place in $n$ independent subspaces without transitions between them, the $n$ different traces over these subspaces are conserved separately Such a separation occurs in case conserved symmetries split the Hilbert space, for instance the total spin is conserved in the dynamics given by (6) if all couplings are equal. Then, the above argument implies the existence of $n$ different eigenoperators with eigenvalue 1. Hence the degeneracy is (at least) $n$ which proves property 1. in the main text.

**Properties 2. and 3.**   As for property 2, we consider an eigenoperator $C$ of $M$ with eigenvalue $\lambda \neq 1$ so that $MC = \lambda C$. Then

$$\mathrm{Tr}(C) = \mathrm{Tr}(MC) \ = \ \lambda \mathrm{Tr}(C) \tag{40}$$

implies $\mathrm{Tr}(C) = 0$, i.e., tracelessness as stated. Since all density matrices can be written as linear combinations of eigenoperators there must be at least one eigenoperator with finite

trace. In view of property 2., this needs to be an eigenoperator with eigenvalue 1 proving property 3. The latter conclusion holds true if we assume that $M$ cannot be diagonalized, but only has a Jordan normal form. If $d_J$ is the dimension of the largest Jordan block, the density matrix $M^{d_J-1}\rho$ will be a linear combination of eigenoperators while still having the trace 1.

**Property 4.** Next, we show that no eigenvalue $\lambda$ can be larger than 1 in absolute value. The idea of the derivation is that the iterated application of $M$ to the eigenoperator belonging to $|\lambda| > 1$ would make this term grow exponentially $\propto |\lambda|^n$ beyond any bound which cannot be true. The formal proof is a bit intricate.

First, we state that for any two density matrices $\rho$ and $\rho'$ their scalar product is non-negative $0 \leq (\rho|\rho')$ because it can be viewed as expectation value of one of them with respect to the other and both are positive operators. In addition, the Cauchy-Schwarz inequality implies

$$0 \leq (\rho|\rho') \leq \sqrt{(\rho|\rho)(\rho'|\rho')} = \sqrt{\text{Tr}(\rho^2)\text{Tr}((\rho')^2)} \leq 1. \tag{41}$$

Let $C$ be the eigenoperator of $M^\dagger$ belonging to $\lambda$; it may be represented as in (36) and scaled such that the maximum of the traces of the $p_i$ is 1. Without loss of generality this is the case for $p_1$, i.e., $\text{Tr}(p_1) = 1$. Otherwise, $C$ is simply rescaled: by $C \to -C$ to switch $p_2$ to $p_1$, by $C \to -iC$ to switch $p_3$ to $p_1$, or by $C \to iC$ to switch $p_4$ to $p_1$. On the one hand, we have for any density matrix $\rho_n$

$$|(C|\rho_n)| \leq |\Re(C|\rho_n)| + |\Im(C|\rho_n)| \leq 2 \tag{42}$$

where the last inequality results form (41). On the other hand, we set $\rho_n := M^n p_1$ and obtain

$$2 \geq |(C|\rho_n)| = |((M^\dagger)^n C|p_1)| = |\lambda^*|^n |(C|p_1)| = |\lambda|^n \sqrt{(\Re(C|p_1))^2 + (\Im(C|p_1))^2} \tag{43a}$$
$$\geq |\lambda|^n |\Re(C|p_1)| = |\lambda|^n (p_1|p_1) \tag{43b}$$

where we used $(p_1|p_2) = 0$ in the last step; this holds because $p_1$ and $p_2$ result from the same diagonalization, but refer to eigenspaces with eigenvalues of different sign. In essence we derived

$$2 \geq |\lambda|^n (p_1|p_1) \tag{44}$$

which clearly implies a contradiction for $n \to \infty$ because the right hand side increases to infinity for $|\lambda| > 1$. Hence there cannot be eigenvalues with modulus larger than 1.

**Property 5.** The matrix $M$ can be represented with respect to a basis of the Krylov space spanned by the operators

$$\rho_n := M^n \rho_0 \tag{45}$$

where $\rho_0$ is an arbitrary initial density matrix which should contain contributions from all eigenspaces of $M$. For instance, a Gram-Schmidt algorithm applied to the Krylov basis generates an orthonormal basis $\tilde{\rho}_n$. Due to the fact, that all the operators $\rho_n$ from (45) are hermitian density matrices $\tilde{\rho}_n = \tilde{\rho}_n^\dagger$, we know that all overlaps $(\rho_m|\rho_n)$ are real and hence the constructed orthonormal basis $\tilde{\rho}_n$ consists of hermitian operators. Also, all matrix elements $(\rho_m|M\rho_n) = (\rho_m|\rho_{n+1})$ are real so that the resulting representation $\tilde{M}$ is a matrix with real coefficients whence

$$\tilde{M}c = \lambda c \tag{46a}$$

implies

$$\tilde{M}c^* = \lambda^* c^* \tag{46b}$$

by complex conjugation. Here $c$ is a vector of complex numbers $c_n$ which define the corresponding eigenoperators by

$$C = \sum_{n=1}^{D} c_n \tilde{\rho}_n. \tag{47}$$

Thus, $c$ and $c^*$ define $C$ and $C^\dagger$, respectively.

**Property 6.** In view of the real representation $\tilde{M}$ of $M$ with respect to an orthonormal basis of hermitian operators derived in the previous paragraph the determination of the eigenoperators with eigenvalue 1 requires the computation of the kernel of $\tilde{M} - \mathbb{1}_D$. This is a linear algebra problem in $\mathbb{R}^D$ with real solutions which correspond to hermitian operators by means of (47). This shows the stated property 6..

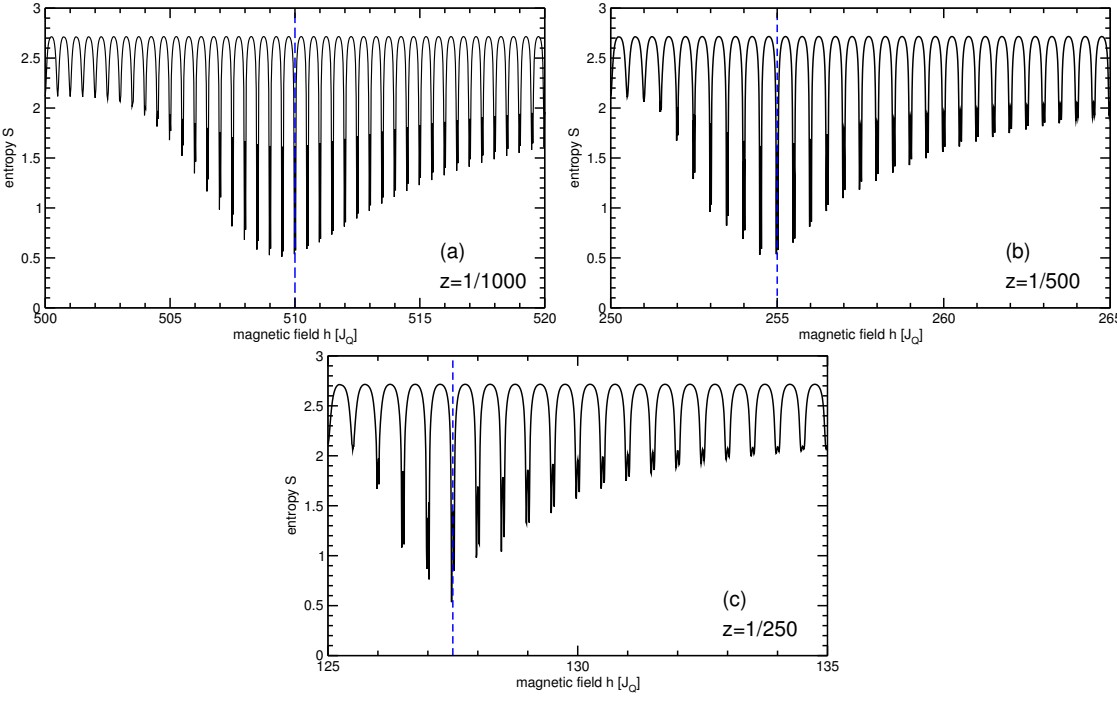

Figure 6: (a) Residual entropy as function of the applied magnetic field for $N = 3$, $J_{\max} = 0.02$, and $z = 1/1000$ to show the position at $h = 2\pi/(zT_{\mathrm{rep}})$ and the shift, dashed line at $\approx 500 J_Q J_{\max}/(2z)$ of the nuclear magnetic resonance. (b) Same as (a) for $z = 1/500$. (c) Same as (a) for $z = 1/250$.

## C  Shift of the Nuclear Resonance

In the main text, the shift of the nuclear resonance due to the coupling of the nuclear spins to the central, electronic spin was shown in the right panel of Fig. 1(a). The effect can be estimated by

$$z\Delta h \approx \pm J_{\max}/2. \tag{48}$$

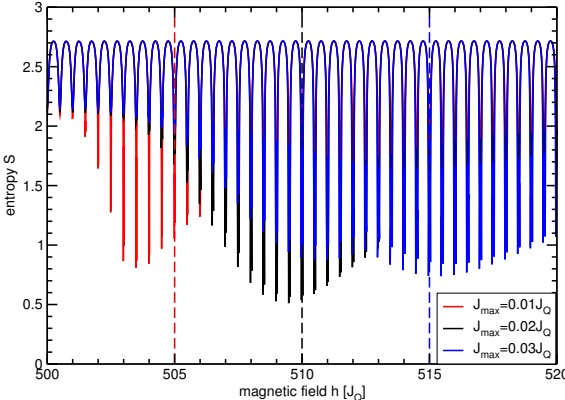

Figure 7: Residual entropy as function of the applied magnetic field for $N = 3, z = 1/1000$ and various values of $J_{\max}$. The shifts indicated by the dashed lines correspond to the estimate (48).

This relation is highly plausible, but it cannot be derived analytically because no indication for a polarization of the central, electronic spin in $x$-direction was found. Yet, the numerical data corroborates the validity of (48).

In Fig. 6, we show that the nuclear resonance without shift occurs for

$$zhT_{\mathrm{rep}} = 2\pi n'$$ (49)

where $n' \in \mathbb{Z}$. But it is obvious that an additional shift occurs which is indeed captured by (48).

In order to support (48) further, we also study various values of $J_{\max}$ in Fig. 7. The estimate (48) captures the main trend of the data, but it is not completely quantitative because the position of the dashed lines relative to the minimum of the envelope of the resonances varies slightly for different values of $J_{\max}$. Hence, a more quantitative explanation is still called for.

# D    Entropy Reduction for Other Distributions of Couplings

In the main text, we analyzed a uniform distribution of couplings, see Eq. (8). In order to underline that our results are generic and not linked to a special distribution, we provide additional results for two distributions which are often considered in literature, namely an exponential parameterization as defined in (9) and a Gaussian parametrization as defined in (10).

The key difference between both parametrizations (9) and (10) is that due to the quadratic argument in (10) the large couplings in this parametrization are very close to each other, in particular for increasing $N$. Hence, one can study whether this feature is favorable of unfavorable for entropy reduction.

Additionally, the difference between $\alpha = 0.5$ and $\alpha = 1$ consists in a different spread of the couplings. For $\alpha = 1$, one has $J_{\min}/J_{\max} = 1/e$ in both parametrizations while one has $J_{\min}/J_{\max} = 1/\sqrt{e}$ for $\alpha = 0.5$, i.e., the spread is smaller.

Figure 8 displays the results for the exponential parametrization (9) while Fig. 9 depicts the results for the Gaussian parametrization (10). Comparing both figures shows that the precise distribution of the couplings does not matter much. Exponential and Gaussian parametrization lead to very similar results. They also strongly ressemble the results

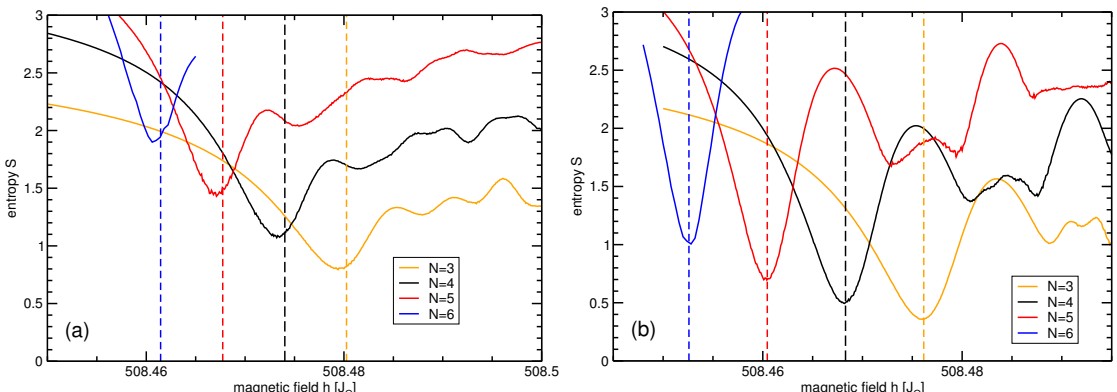

Figure 8: Residual entropy as function of the applied magnetic field for various bath sizes $N$ for the exponentially distributed couplings given by (9); panel (a) for $\alpha = 1$ and panel (b) for $\alpha = 0.5$ and hence smaller ratio $J_{\min}/J_{\max}$.

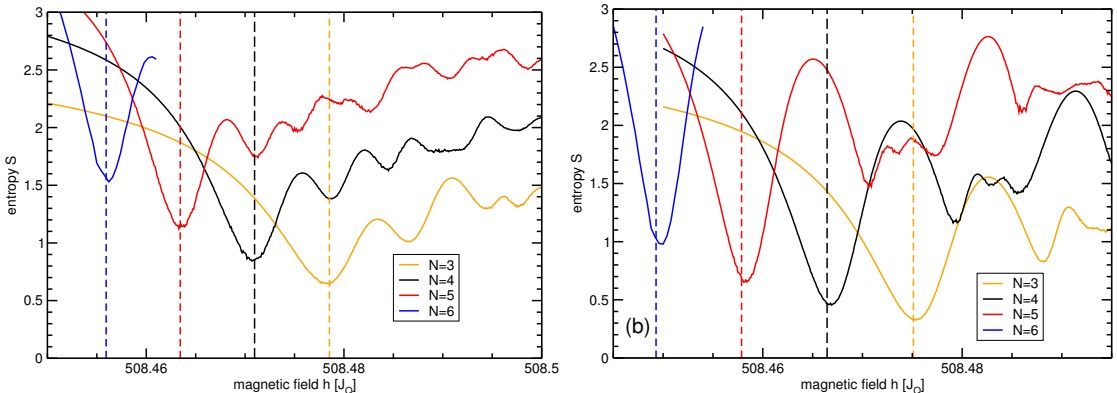

Figure 9: Residual entropy as function of the applied magnetic field for various bath sizes $N$ for the Gaussian distributed couplings given by (10); panel (a) for $\alpha = 1$ and panel (b) for $\alpha = 0.5$ and hence smaller ratio $J_{\min}/J_{\max}$.

shown in Fig. 2a in the main text for a uniform distribution of couplings. This is quite remarkable since the Gaussian parametrization leads to couplings which are very close to each other and to the maximum coupling. This effect does not appear to influence the achievable entropy reduction.

The ratio $J_{\min}/J_{\max}$ between the smallest to the largest coupling appears to have an impact. If it is closer to unity, here for $\alpha = 0.5$, the reduction of entropy works even better than for smaller ratios.

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
