# Peer review of "Quantum Coherence from Commensurate Driving with Laser Pulses and Decay"

_SciPost Physics_

## Round 1 · Referee Report · Anonymous (Referee 1) · 2019-12-29

Report

In the manuscript, the author shows that, with a periodic driving, a quantum system converges to a coherent quantum state if a highly-excited state is involved.
In particular, the author considers a small spin system, which can be studied by diagonalizing computationally the density matrix.

The main message of this work is interesting, and the manuscript is generally well written.
I like the current structure of the paper, but I think that the text can be slightly improved, considering the following remarks.

While the Introduction (section 1) is sufficiently clear and contains enough references, the Model (section 2) can be improved.
For instance, some variables between Eqs. (2) and (3) are not explicitly defined, like z, ot the Bohr magneton \mu_B: I prefer a pedantic description, than a vague one.
Besides this, I think that the model could be explained more in detail, and the whole section should be written to be accessible to a general reader.
Additional references should be added, to properly introduce all the physical entities involved (Overhauser field, trion, Lindblad equation).

The author could briefly discuss the range of validity for the present system of the Master Equation in Lindblad form, for which the Born, Markov, and rotating-wave approximations must hold.

In my black and white copy of the paper it is not easy to distinguish clearly between the different curves in the figures, in particular in Figs. (3,4,5,7,8,9).
The author could use different line widths, dots or dashes, or choose colours that are easier to distinguish in black and white.

If these suggestions are addressed, I recommend to publish the paper.
  • validity: -
  • significance: -
  • originality: -
  • clarity: -
  • formatting: -
  • grammar: -

Author:  Götz Uhrig  on 2020-02-07  [id 733]

(in reply to Report 1 on 2019-12-29)
Category:
remark

Dear Referee,

thank you for your careful reading of my manuscript and the constructive proposals. I appreciate the judgement that the work is "interesting, and the manuscript is generally well written" and that you "like the current structure of the paper". The following improvements have been made:

The Model section has been expanded considerably explaining all symbols and quantities explicitly.

The Lindblad formalism is very well established for the decay of high-energy states (1eV) implying a large phase space for the emitted photon and a very fast dynamics of the bath. Hence the reference to a text book in the revised version, here the one by Breuer and Petruccione, to justify the used equations appears sufficient to me.

Since the publication is planned in an online-journal the use of color in the figures to distinguish different curves is not a caveat in my opinion. So I prefer to stick to this feature because a different line style is used for indicating the positions of resonances and shifts. Using different lines styles also for different bath sizes would be prone to misunderstandings.

---

## Round 1 · Referee Report · Anonymous (Referee 2) · 2020-1-7

Strengths

1) The central result of this paper, the new (to my knowledge) mechanism inducing the entropy reduction and the transition to an ordered regime (due to the commensurability condition), is very nice and interesting. In general, I expect that the analysis developed in this paper should stimulate further experimental and theoretical work.

2) The paper is well organized. After briefly reviewing the basic information about the model and the mathematical aspects of the evolution protocol, it focuses the reader attention on the significant results about the entropy properties and the distillation process.

Weaknesses

1) The readability of this paper must be improved. To this end various unclear points or comments should be modified and several errors should be removed. These are mainly concentrated in Section 4, the most important part of this paper. This section also features a very technical style and many comments are made (or formulas are used) with the implicit assumption that any reader can understand them. The author does not provide the information necessary to understand (at least, in general) many aspects and intermediate steps of the discussion made in section 4.

Report

This paper investigates the effect of periodic pulse pumping on a spin system in which the energy-exchange processes are affected by the presence of an intermediate highly-excited decaying state. The latter is represented by a high-energy trion-state term which forms the model Hamiltonian together with the electron-spin, nuclear-spin, and spin-spin interaction terms. The presence of the trion state combined with the commensurability of periodic pulses (to the characteristic time of internal processes) allows one to show how long trains of pulses enact a distillation process leading to a “coherent” quantum state in which the disorder of the initial state is essentially suppressed. The evolution of the density matrix describing the system dynamics is governed by a Lindblad equation in which a standard damping term takes into account the dissipative effects relevant to the trion-state decay. Numerical simulations also allow to determine the entropy properties of the final “coherent” state and, particularly, to highlight the entropy reduction triggered by the combination of pulse trains and dissipative effects in the asymptotic quasi-stationary regime.

The paper is well organized. After briefly reviewing the basic information about the model Hamiltonian and the mathematical aspects of the evolution protocol (sections 2 and 3), focuses the reader attention on i) the entropy properties and ii) the dependence of the distillation process from the number of pulses. These aspects are extensively discussed in sections 4 and 5. Further details about important but technical aspects of sections 2 and 3 are given in three final appendices which concern 1) the linear mapping representing the pulsed dynamics in the Lindblad picture, 2) the mathematical derivation of the properties stated in section 3, and 3) the entropy for different distributions of the Overhauser-field hyperfine interactions. The results presented in this paper are, in general, technically sound. The system dynamics is studied by using the well-established Lindblad formalism and is supported by a non trivial but detailed discussion on the density-matrix linear mapping, the core mechanism of the evolution. The central result of this paper, the new (to my knowledge) mechanism inducing the entropy reduction and the transition to an ordered regime (due to the commensurability condition), is very nice and interesting. In general, I expect that the analysis developed in this paper should stimulate further experimental and theoretical work. A revision, however, is necessary to improve the readability of this paper. To this end various unclear points or comments should be modified and several errors should be removed. These are mainly concentrated in Section 4, the most important part of this paper. This section also features a very technical style and many comments are made (or formulas are used) with the implicit assumption that any reader can understand them. The author does not provide the information necessary to understand (at least, in general) many aspects and intermediate steps of the discussion made in section 4. Useless to say that supporting comments and formulas by introducing many citations is not sufficient to compensate the absence of clarity.

Comments 1) Formula 5. The definition of operator U puls is not clear as well as the terms “istantaneous” and “unitary”. Since U puls is a hermitian operator the use of term “unitary” should be justified and its role in describing a laser pulse should be explained. 2) At page 4, the origin of the distribution of couplings J i (Overhauser-field distribution) should be briefly discussed and the (physical) reason why one can consider different distributions (within the current model) should be explained. 3) page 5, lines 5-6: if the time unit is h̄/J q , then the trion decay rate should be 2γ = 2.5 J Q /h̄ while the trion life time should be 0.4 ns instead of 0.4 ps. 4) The author considers distributions (8), (9) and (10) which, apparently, are constructed in an arbitrary way. As noted above, some information about this freedom could improve the clarity of this paper. Also, I cannot understand why these distributions depend on J max but are independent from J min . 5) Page 5. The comment below eq. (11) “The commensurability of these resonances is crucial for the advocated mechanism” must be improved. A similar comment is already present in the Introduction and the commensurability condition is again mentioned below eq. (8). The author does not state with the necessary clarity what are the internal processes (and the relevant characteristic times) which allow one to define the commensurability condition. Even if this is very obvious to the physicists of this specific research field, I am afraid that it is not so obvious to non expert readers. 6) A factor h̄ is missing in hT rep = 2πn and zhT rep = 2πn 0 . 7) The author should better explain how the two formulas defining the Larmor-precession resonances hT rep = 2πn and zhT rep = 2πn 0 (defined after formula (10)) are related to the periodic maxima (or minima) of S in fig. 1. Two nested resonances are apparently discernible in fig. 1 (see the relevant comment in the text). However, after stating the presence of such resonances, the author apparently contradicts this claim observing that these conditions are not applicable without pulsing. It is hard to understand this point. 8) The caption of Fig. 1 states that “Resonances of the electronic spin occur every ∆h = 0.5J max ”. However, looking at the figure and observing that the h units are J Q , the separation between subsequent electron-spin resonances seems to be ∆h = 0.5J Q . Concerning the resonances of nuclear spins these should occur every ∆h = 500J max = 10J Q : figure 1a (right panel) suggests that the shift is apparently 10 units, if this is referred to the value h = 500J Q . However, why this value should be the reference value is not very clear. The blue dashed line depicts the offset from the nuclear resonance (placed at h = 509.5J Q ) defined by ∆h = ±J max /(2z) = ±500J max = ±10J Q , but figure 1b (left panel) apparently shows that the offset is ∆h = 0.5J Q . Concluding, it seems really hard to identify the shifts mentioned in the comment “The driven systems displays important shifts”. The caption of fig. 1 is very confused and the corresponding comment in the text does not clarify the rich information encoded in it. Due to its relevance, this part should be carefully checked and its clarity significantly improved. 9) The comment (see below eq. (11)) “where A max is the maximum Overhauser field” should be “where A max is the maximum value of the Overhauser field”. 10) Page 9: To achieve a satisfactory understanding of the discussion concerning figure 4 (and of the benefits due to the spread reduction) is obviously related to the referee comment 4 about the alternative parametrization of J i .

Requested changes

1) see the file in attachment or the report

Attachment

  • validity: high
  • significance: high
  • originality: high
  • clarity: low
  • formatting: excellent
  • grammar: excellent

Author:  Götz Uhrig  on 2020-02-07  [id 732]

(in reply to Report 3 on 2020-01-07)

Dear Referee,

thank you for the very thorough reading of the manuscript and the constructive suggestions. I appreciate very much that you find the submitted work "very nice and interesting" and prone to "stimulate further experimental and theoretical work." Concerning the readability, I improved the manuscript along your suggestions:

ad 1)
The description of U_puls has been expanded and its unitarity is stated explicitly. Its hermiticity is actually accidental. The key feature is that it describes the transition from the electronic ground state to the excited trion state and viceversa.

ad 2)
The origin of the couplings as hyperfine couplings between the electron spin and the nuclear spin has been explained.

ad 3)
\hbar is actually set to unity. This is recalled in the revised version and the notation is made consistent with this convention. The error in the units has been corrected (ps->ns).

ad 4)
The distributions are motivated by the shapes of wave functions as is explained in the revised version. Still, of course, there is some degree of arbitrariness in this proof-of-principle study. The value of J_min is implicit in the parametrizations for given parameter alpha because J_max sets the energy scale. This is clarified in the revised version.

ad 5)
The statements on commensurability have been expanded significantly. It is clearly stated what has to be commensurate with what.

ad 6)
Since \hbar is set to unity it is not missing in the formulae.

ad 7)
I agree that the statements were unclear and misleading. They have been rephrased and extended in the revised version. The conditions are applicable and even require the pulsing. But the coupling of the spins leads to the shifts that are discussed. Thank you for pointing out the previous deficiency of the presentation.

ad 8)
There was an error in the caption. The energies and field are indeed given in units of J_Q, not J_max. The present version is certainly clearer.

ad 9)
The suggested formulation has been adopted.

ad 10)
The discussion of "spread" has been expanded so that the argument is clearer now.

---

## Round 1 · Referee Report · Anonymous (Referee 3) · 2020-1-8

Strengths

Good style, appropriate length, topical problem

Weaknesses

The main mechanism of the suppression of non-wanted states could be better worked out and explained to non-experts in the fields of open quantum systems and spin problems.

Report

The contents and general style of the paper is good. I guess it deserves publication. The "main mechanism" that is repeated several times between the introduction, main part and conclusion could be better explained, however, and if possible in more physical terms referring to precise formulae. Then, I list some minor comments which should be addressed by the author before acceptation:

1) the requested number of pulses seems quite unrealistic as the author himself writes in the paper: is there no way around this deficit, e.g. by creating more intelligent pulses or drivings (see quantum control theory)?

2) if the nonintersecting states are "gradually suppressed" as written e.g. in the conclusion, what is then the norm of the remaining states relative to the initial norm? Or is all the norm pumped into the remaining states (but then there is no real dissipation present, just decoherence). What would happen in a truly open system with norm decay to some "external" level/state? Would the mechanism still work?

3) Next to the two paper refs. 2 and 4 I suggest a simultaneous work proving the possible increase of coherence by dissipation: Phys. Rev. Lett. 101, 200402 (2008).

Requested changes

1) the requested number of pulses seems quite unrealistic as the author himself writes in the paper: is there no way around this deficit, e.g. by creating more intelligent pulses or drivings (see quantum control)?

2) if the nonintersecting states are "gradually suppressed" as written e.g. in the conclusion, what is then the norm of the remaining states relative to the initial norm? Or is all the norm pumped into the remaining states (but then there is no real dissipation present, just decoherence).

3) Next to the two paper refs. 2 and 4 I suggest a simultaneous work proving the increase of coherence by dissipation: Phys. Rev. Lett. 101, 200402 (2008).

  • validity: high
  • significance: high
  • originality: good
  • clarity: ok
  • formatting: good
  • grammar: good

Author:  Götz Uhrig  on 2020-02-07  [id 731]

(in reply to Report 2 on 2020-01-08)
Category:
remark
pointer to related literature

Dear Referee,

thank you for your careful reading of the manuscript and the constructive comments.

ad 1)
It is an interesting idea to optimize the periodic driving by choosing special cycles of
pulses or by shaping the pulses. I formulated this in the revised manuscript as a promising route
of future research. The present article focusses on establishing the fundamental phenomenon.

ad 2)
In the revised Conclusion I explain that indeed weight is shifted from "suppressed" states to "enhanced" states. The enhanced ones are those with internal dynamics commensurate to the periodic driving. Note that the trace of the density matrices is conserved upon application of the dynamic matrix, but other norms are not. It must be stressed that the system truly is a dissipative open system. Otherwise, no changes of weight would be possible at all.

ad 3)
The suggested reference is included.

---

## Editorial Decision

resubmitted